# Decoding kinase-adverse event associations for small molecule kinase inhibitors

Xiajing Gong[1], Meng Hu[1], Jinzhong Liu[1], Geoffrey Kim[2], James Xu[3], Amy McKee[4], Todd Palmby[2], R. Angelo de Claro[1] & Liang Zhao [1] ✉

Small molecule kinase inhibitors (SMKIs) are being approved at a fast pace under expedited programs for anticancer treatment. In this study, we construct a multi-domain dataset from a total of 4638 patients in the registrational trials of 16 FDA-approved SMKIs and employ a machine-learning model to examine the relationships between kinase targets and adverse events (AEs). Internal and external (datasets from two independent SMKIs) validations have been conducted to verify the usefulness of the established model. We systematically evaluate the potential associations between 442 kinases with 2145 AEs and made publicly accessible an interactive web application "Identification of Kinase-Specific Signal" (https://gongj.shinyapps.io/ml4ki). The developed model (1) provides a platform for experimentalists to identify and verify undiscovered KI-AE pairs, (2) serves as a precision-medicine tool to mitigate individual patient safety risks by forecasting clinical safety signals and (3) can function as a modern drug development tool to screen and compare SMKI target therapies from the safety perspective.

As more genetic drivers of disease progression are discovered, small molecule kinase inhibitors (SMKIs) are becoming a rapidly expanding class of oral drug products that have been demonstrated to be efficacious in the targeted therapy for various malignancies[1,2]. Imatinib was the first SMKI introduced into clinical oncology in the early 2000s. Thereafter, the Food & Drug Administration (FDA) has approved a total of 65 SMKIs including gefitinib, erlotinib, and sorafenib by 2020. As SMKIs have become standard therapies for many cancers and other life-threatening diseases, information on treatment-emergent adverse events (TEAEs) becomes abundantly available[1,3,4]. Early and prompt characterization of potential SMKI-related adverse effects (AEs) in the drug development process can critically improve research and development (R&D) efficiency and reduce product attrition rate, especially for oncology products through expedited programs.

Although SMKIs share similar mechanisms of action, they differ in the spectrum of targeted kinases, pharmacokinetics (PK), and substance-specific AE profiles. Both on-target and off-target kinase inhibitions are potentially responsible for AEs. For example, all

approved SMKIs (e.g., pazopanib, sunitinib, and regorafenib) primarily targeting vascular endothelial growth factor receptor 2 (VEGFR2) share warnings of hypertension and bleeding[5] (i.e., "on-target" side effect); however, these SMKIs also exhibit heterogenous kinase inhibition (KI) profiles beyond VEGFR2 inhibition, resulting in a broad range of toxicity and side effects in a variety of tissues and organs (i.e., "off-target" side effect). Empirical evidence from clinical practice tends to attribute AEs to the targeted KI of an SMKI but underestimates the off-target effects[6]. Various attempts have been made to discover other hidden potential associations between KI and drug toxicity via conventional statistical approaches including bioinformatics methods[7–10] using uni-dimensional input data at the genome level or molecular level. However, these practices considered AEs curated from literature or databases such as FAERS (the FDA adverse event reporting system) at the population level, and were limited by access to patient-level clinical data in terms of drug exposure and clinical responses. In addition, considerable interindividual variabilities in drug exposure are present for SMKIs, and research has shown causal relationships between the degree of drug exposure and AEs[11]. Hence, previous KI–AE association

[1]Center for Drug Evaluation and Research, Food and Drug Administration, Silver Spring, MD, USA. [2]BeiGene, Cambridge, MA, USA. [3]Potomac Oncology and Hematology, Rockville, MD, USA. [4]Parexel, Washington, DC, USA. ✉e-mail: Liang.Zhao@fda.hhs.gov

studies did not consider individualized drug exposure–AE response relationships, making it challenging to use the characterized KI–AE relationships to predict personalized TEAEs of an SMKI based on individual drug exposure and KIs.

Time-to-event analysis (i.e., survival analysis) is commonly used in the analysis of the time to occurrence of a specific event including TEAEs[12]. The regression-based survival method, the Cox proportional hazard regression model, is a commonly used approach to estimate the probability of an event occurrence at a certain time[13]. However, this model relies on certain assumptions, i.e., the coefficients of the predictor variables are constant over time and their effects are additive on one scale. Recently, a well-established survival analysis approach, the random survival forest (RSF) method, was developed as a machine learning (ML) approach[14]. Importantly, leveraging the data-adaptive nature of ML approaches, the RSF method does not have prior assumptions on the relationship to be characterized. It also offers superior performance to analyze high-order, high-dimension, and nonlinear relationship survival data. Our recent simulation study has demonstrated that RSF adeptly accounts for nonlinearity, correlation, and interaction of the predictor variables[15]. Therefore, RSF will be an ideal method for the predictive analysis of time to AE data involving both continuous and categorical data types as inputs and heterogenous relationships among all variables including potential kinase-kinase interactions.

The 65 SMKIs approved by the FDA provide a wealth of data that are untapped for a pooled analysis to unravel the undiscovered relationships behind KI and clinical AE. In addition, computational advances allow us to use population PK models to derive SMKI systemic exposures at the patient level. Consequently, we can construct a multi-domain, heterogeneous, and high-dimensional dataset integrating patient-level demographic, PK, and safety data, as well as nonclinical information such as in vitro KI potency profiles of SMKIs. The RSF-based ML approach allows the establishment of an actionable model to predict individualized AEs de novo. The established approach will provide significant insights into the safety evaluation and prediction for both on-target and off-target effects of the FDA-approved SMKIs, and illustrate a proof-of-concept to harness ML and quantitative systems pharmacology for prospective identification of personalized kinase-specific safety signals.

## Results

### Review of AEs following SMKI treatments
Out of the 65 SMKIs approved by the FDA up to December 31, 2020 (Fig. 1a), 16 of them in their registrational studies had patient-level PK and AE data with available in vitro KI profiling data (the dissociation constants $K_d$[16]). A total of 4638 patients from the 16 pivotal studies were included in the analysis (Supplementary Table 1). The associations between SMKIs and AEs were integrated for the 16 pivotal studies ("Methods"). Figure 1b showed an example of vandetanib, afatinib, erlotinib, and nintedanib, four SMKIs all targeting EGFR. We pooled the top ten AEs associated with each SMKI and computed their reporting odds ratio (ROR) to indicate the relative risk of the AEs. The four EGFR inhibitors shared a similar risk of on-target AEs of decreased appetite, nausea, diarrhea, fatigue, and vomiting, but exhibited overall heterogeneous AE profiles, suggesting varying off-target effects.

### ML modeling for KI-AE associations
The RSF-based ML method was implemented to predict clinically observed AEs following SMKI treatments. The workflow of data integration and predictive modeling were illustrated in Fig. 1c ("Methods"). We employed the ML model to identify pairs of KI–AE associations based on the full integrated database. Subsequently, variable importance (VIMP) was applied to assess the impact of an input variable on the model predictive performance. A large value indicates a strong association between the interested KI and AE, whereas a small or negative value indicates low or no association. The top 25 kinases for each of the 5 representative AEs were shown based on their VIMP values (Fig. 2a). Similar results were obtained using the minimal depth approach (Supplementary Fig. 2).

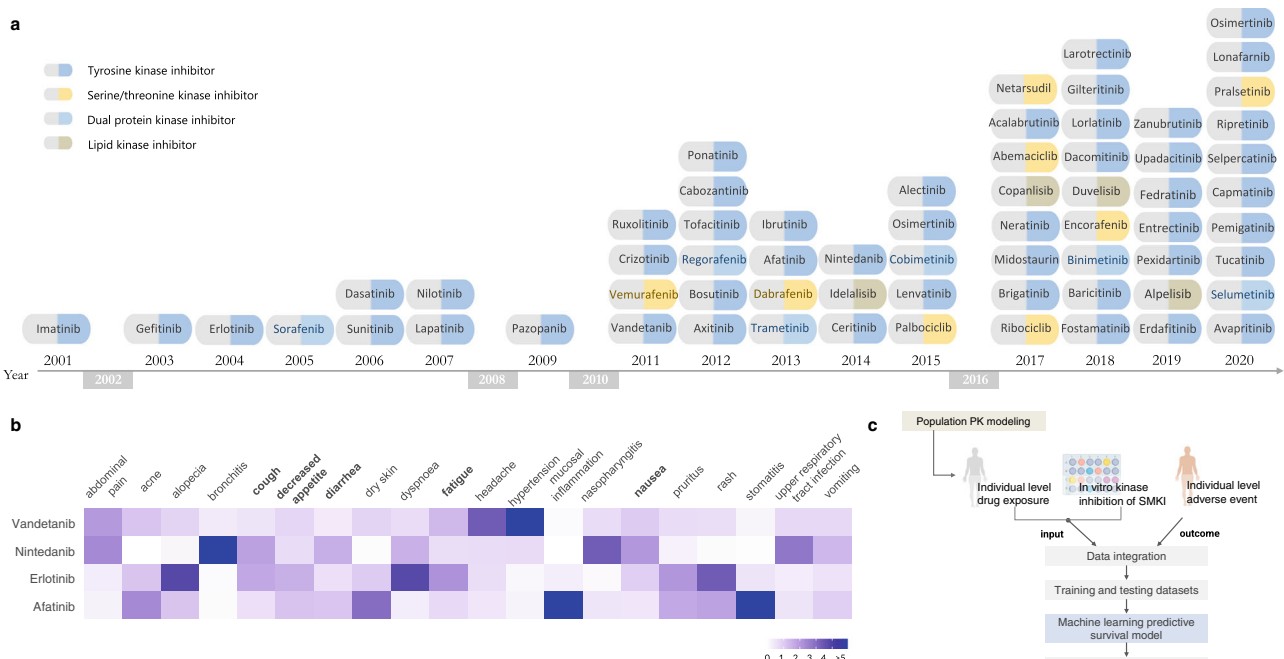

**Fig. 1 | ML-based model was developed based on clinical and non-clinical data of FDA-approved SMKIs to prospectively identify personalized kinase-specific safety signals. a** Small molecule kinase inhibitors (SMKIs) approved by the U.S. FDA up to December 31, 2020. SMKIs were listed in the chronological order of their first FDA approval date with their generic names. Source data were retrieved from the FDA public website. **b** Characteristics of reporting odds ratio (ROR) profiles for combined top ten AEs of vandetanib, afatinib, erlotinib, and nintedanib. **c** The workflow of data integration and predictive modeling.

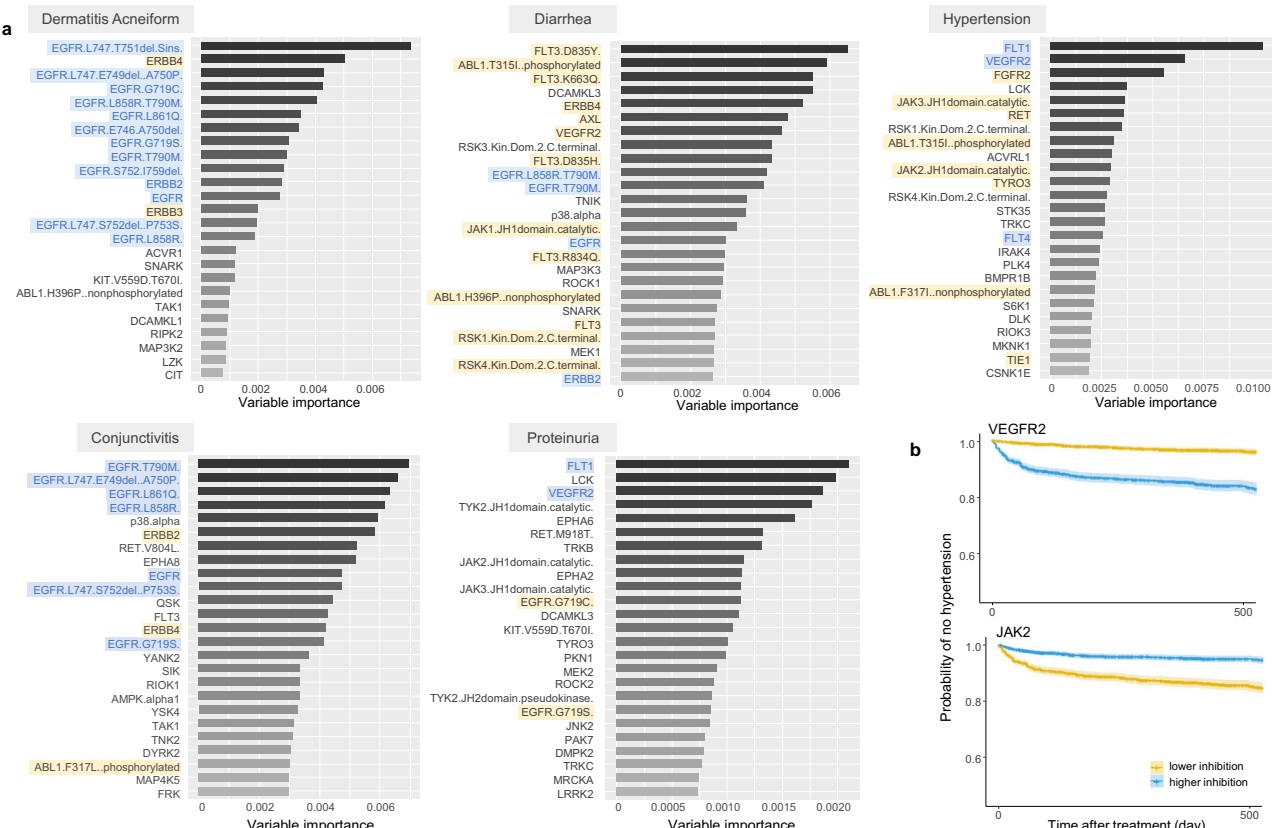

**Fig. 2 | Potential association between kinase and AE as identified by ML modeling. a** Variable importance (VIMP) assessment of *n* = 444 predictive variables for five representative AEs. Bar length indicates the VIMP value of the ensemble model, which represents the difference in the out-of-bag model prediction errors with and without this predictive variable being permuted[43]. The identified top 25 predictive variables are listed for each AE. The blue-highlighted kinases are the representative experimentally well-established KI–AE pairs[6]. The yellow-highlighted pairs are validated by KI–AE pairs found in literature survey results for KI–AE pairs post the[6] publication review (for references, see Supplementary Table 3). **b** Kaplan–Meier

survival (no hypertension) probability curves stratified by VEGFR2 (top) and JAK2 (bottom) inhibitions based on individual-level drug exposures. Each point on the curve is the average of the ensemble survival function over all patients for a given time, with the error bands showing 95% pointwise confidence intervals for all patients together. The divergence between the two survival curves showed that the patients with "higher inhibition" (>median) on VEGFR2 had earlier hypertension onset than those with "lower inhibition" (<median), while the inhibition on JAK2 showed the opposite outcome.

As part of model validation, our results show that based on the computed VIMP values (Fig. 2a, blue highlight), the model-identified KI–AE associations successfully covered the experimentally well-established ones[6] including VEGFR–hypertension, EGFR/ERBB2–diarrhea/dermatitis acneiform, EGFR–conjunctivitis, and VEGFR–proteinuria. Of note, ref. 6 only summarized the reported findings by 2013. As such, we conducted an additional comprehensive survey for literature published after 2013 on kinase-associated toxicities and examined the findings against the mode-identified KI-AE pairs. The literature survey process and results are described in the Supplementary Information section "Literature survey on associations between kinase targets and AEs." More predicated KI-AE pairs were found to be validated by the published experimental evidence (Fig. 2a, yellow highlight). For instance, for diarrhea, besides its well-established association with EGFR, the model has also identified its reported association with other kinases such as FLT3, VEGFR2, and AXL[17]. The VIMP results identified other potential associations that will not be able to be confirmed until new clinical evidence is published, e.g., the association between hypertension and JAK family kinases including JAK2 and JAK3. For this case, we generated Kaplan Meier survival probability curves with patients stratified into higher and lower inhibition groups against VEGFR2 and JAK2 based on their drug exposures (Fig. 2b). The results showed that patients with higher inhibition (>median) on VEGFR2 had earlier hypertension onset than those with lower inhibition (<median),

consistent with the observation in clinical practice. Interestingly, inhibition on JAK2 showed the opposite outcome. Of note, RSF-based ML approach showed non-inferior prediction performance when compared to other ML approaches (please see Supplementary Information section "Evaluation of AE prediction performance using DeepHit and ANN").

**Evaluation of AE prediction performance**
To evaluate the predictive performance of the ML-based method on the patient population level, bootstrapping cross-validation was applied to the integrated patient population dataset. Five representative AEs that affect various organ systems were selected for the evaluation, including diarrhea (gastrointestinal disorders), dermatitis acneiform (skin and subcutaneous tissue disorders), hypertension (vascular disorders), conjunctivitis (ocular disorders) and proteinuria (renal and urinary disorders). For each AE-type evaluation, 80% of the dataset was bootstrap sampled as the training dataset, and the remaining 20% of the dataset was used for model validation. The predictive performance as measured by C-index and 90% confidence interval are shown in Table 1. The results showed that the ML model provided reasonably accurate prediction for AEs occurring either at a lower frequency such as proteinuria (1.5%) or at a higher frequency such as diarrhea (41.6%), as well as for AEs with either shorter onset time such as dermatitis acneiform (14 days) or longer onset time such as conjunctivitis (104 days).

In the clinical settings, more concerns are given to severe AE with grade >3 because they are often associated with a high rate of therapy discontinuation or modification of dosing regimens. The most common grade 4 AEs were thrombocytopenia, neutropenia, leukopenia, lymphopenia, and decreased hemoglobin. The C-index values based on the bootstrapping cross-validation results were 0.908, 0.920, 0.887, 0.912, and 0.864 for the above five grade 4 AEs (Table 2). Overall, a C-index of 0.776 was obtained in predicting grade

4 (life-threatening) AEs, and a C-index of 0.724 was obtained in predicting grade 5 AE (death).

In addition, as one application of the developed model is to predict TEAEs of interested SMKIs with known KI profiles at patient level, leave-one-out cross-validation method was applied to assess the patient level predictive performance. Overall, a mean C-index value of 0.701 was obtained with the leave-one-out cross-validation, indicating that the ML model provided good prediction at the individual level.

### External validation on independent SMKI datasets

Data from 116 and 136 patients with breast cancer in two neratinib monotherapy studies, as well as 349 patients with chronic myelogenous leukemia in an imatinib monotherapy study, were selected as independent datasets for external model validation. We applied the previously trained ML model to predict patient level AE probabilities for these studies and averaged across the study population. Using 10% incidence as a threshold (the reported incidence in ≥10% subjects was used as criterion for "frequently-reported TEAE" in the clinical study reports of neratinib and imatinib), the model predicted AEs with high probabilities cover all the frequently reported TEAEs in neratinib studies, and the majority of frequently reported TEAEs in imatinib study, indicating good model sensitivity. While there are model-predicted AEs with ≥10% probability that was observed for <10% of patients, the model specificity is not compromised considering thousands of AEs are true negatives with predicted probabilities of <10%. Particularly, the specificity is above 95% for all three datasets. Figure 3 indicates that the most frequently reported (in ≥10% of patients) TEAEs are generally well captured by ML models with AE probability predicted to be ≥10%, except for four TEAEs in the imatinib study. Among these four TEAEs, for "dyspepsia" and "hypophosphataemia", the ML model-predicted probabilities fall

### Table 1 | Predictive performances of the machine learning model on patient population data assessed by C-index

| AE (patients affected, %) | Time to AE onset[a] median [days] (range) | C-index (90% confidence interval) |
|---|---|---|
| Diarrhea (41.6) | 25 (1, 616) | 0.712 (0.689, 0.735) |
| Dermatitis acneiform (2.4) | 14 (1,337) | 0.852 (0.779, 0.912) |
| Hypertension (9.3) | 57 (1, 663) | 0.782 (0.751, 0.809) |
| Conjunctivitis (2.2) | 104 (3, 672) | 0.584 (0.510, 0.638) |
| Proteinuria (1.5) | 58 (1, 591) | 0.784 (0.696, 0.868) |

[a]Distribution of time to event for each AE dataset is displayed in Supplementary Fig. 5.

### Table 2 | Predictive accuracy for grade 4 life-threatening AEs

| Top five grade 4 AE | Patient affected (%) | C-index (median) |
|---|---|---|
| Thrombocytopenia | 5.0 | 0.908 |
| Neutropenia | 4.3 | 0.920 |
| Leukopenia | 1.2 | 0.887 |
| Lymphopenia | 0.9 | 0.912 |
| Hemoglobin decreased | 0.7 | 0.864 |

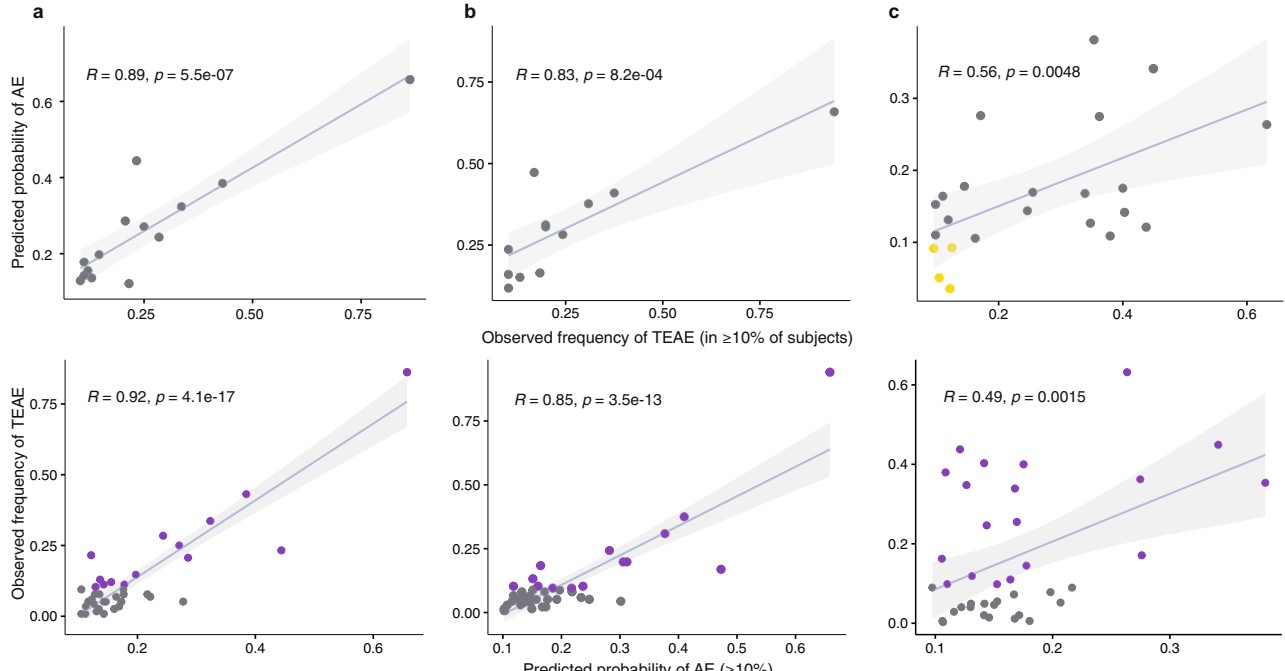

**Fig. 3 | External validation on independent SMKI datasets: relationships between the model predicted AE probabilities and the observed AE incidences.** Columns **a**–**c** are results based on two neratinib and one imatinib studies, respectively. The top panels show the scatterplots for the corresponding predicted probabilities vs all the frequently reported TEAEs (in ≥10% of study subjects) and the bottom panels for the corresponding observed TEAEs vs all AEs predicted to occur in ≥10% of study subjects. On each plot, the blue line represents the fitted linear regression line, and the gray band represents the 95% confidence interval; Pearson's r (two-tailed) with p value is displayed. Yellow points represent four frequently reported TEAEs with predicted probabilities of <0.1; purple points AEs with both predicted probabilities and observed incidences of ≥10%.

marginally below the threshold 10%; for "dysgeusia" and "weight increased", it is noticed that event occurrence rates of these two AEs are 5 and 2%, respectively, in the training dataset used for building ML models. This level of sparsity in data could compromise the ML model performance, due to the high data dependency of ML approaches. Overall, the model-predicted AE probabilities are significantly correlated with the observed frequencies for all three datasets. We also conducted analyses using two higher threshold values (i.e., 15 and 20%) and consistent results were observed (Supplementary Fig. 3).

As an additional validation, we compared the predictions from ML models with AEs reported in the FDA Adverse Event Reporting System (FAERS) regarding neratinib and imatinib. The validation process and results are described in the Supplementary Information section "Comparison of the predictions from ML models with AEs reported in FAERS." Overall, the majority of top FAERS-reported AEs were captured by the ML model with predicted high probability. Of note, there are several AEs specific to pharmacovigilance or AE with preferred terms not reported in clinical trials (e.g., "off label use" in Supplementary Table 5)—since the predictive ML model was developed based on data from clinical trials, there is no model and prediction for those AEs.

### Interactive web application for identification of personalized kinase-specific safety signal

In this study, the validated ML method systematically evaluated the potential associations for a total of 948,090 KI–AE pairs combining 442 kinases with 2145 AEs in customized preferred terms. To present the daunting amount of learned information, we established a non-commercial and interactive web application "Identification of Kinase-Specific Signal" (https://gongj.shinyapps.io/ml4ki) that the users can utilize to query, retrieve, analyze, and visualize kinase–AE or AE–kinase associations. The web application can support (1) queries of model-identified KI–AE associations by searching a particular AE or a particular kinase, (2) prediction of personalized TEAE for a compound of interest based on the KI profiling and drug exposure level, where the compound can be either an FDA-approved SMKI or a new compound in development.

## Discussion

In this study, leveraging patient-level demographic information and drug exposure data, off-target and on-target AEs, we employed ML method to prospectively identify personalized kinase-specific safety signals. The credibility of the modeling results have been established by its capability to (1) effectively identify the significant correlations between KIs and AEs, some of which were validated by well-established KI–AE associations in the clinic and/or literature; (2) predict grade 4 and 5 AE occurrences at the population level with a mean C-index of 0.776 and 0.724, respectively; (3) predict the AE occurrence at the patient level with a mean C-index of 0.701; and (4) demonstrate a significant correlation between the model predicted AE risk and the observed safety profile of TEAE with independent datasets from two external SMKIs.

The high attrition rate of drug development across the pharmaceutical industry has been well documented[18], and attributed to undesired safety profiles[19]. Leveraging big data methodologies represents not only one of the 21st century modernization initiatives for drug R&D but also an important part of personalized medicine[20]. Big data technology and data science can transform huge, heterogeneous, and dynamic biological and clinical data into interpretable and actionable models. Numerous computational methods have been developed to predict AEs based on chemical structure, biological activity and spectra, binding profiles of protein–ligand, and phenotypic information[7,8,21,22]. However, these methods have not taken patient-level drug exposures into account[11]. In particular, Yang and

colleagues integrated the information of AE frequency for SMKIs and SMKIs inhibition for KTs (kinase targets) in terms of $K_d$ from literature mining to generate an AE–KT association score matrix, where the higher score value indicates the more prioritized AE–KT association[23]. Our previous work[24,25] established a similar matrix of AE–kinase pairs and applied disproportionality analyses to identify significant KI–AE associations. Federer and colleagues focused on the identification of the linkage between drug and AE rather than the kinase–AE pairs[26]. It is worth noting that our current study integrated published in vitro kinome data ($K_d$) similar to what was carried out by Yang[23], although with increased kinase targets and access to patient-level clinical data. In general, the previous works cannot offer a framework to support AE prediction at an individual subject level as in our current study. Specifically, the analysis data integrated multi-domain variables including in vitro kinase inhibitory profiles of the tested substances, patient demographics, and individual drug exposure to predict the time and occurrence of AEs. The patient-level PK data make it feasible to account for a portion of interindividual variability in clinical safety response towards the same dose. The multi-domain information, from the molecular level, patient demographic level, and individual PK level as a result of in vivo drug–human pharmacological interaction, has led to satisfactory model performance in terms of positively capturing all known KI–AE pairs and predicting individual AE occurrences with a mean C-index of 0.701.

The ML-based model can be readily scaled up to incorporate more information such as in vitro kinome and pharmacogenomic data to improve its predictive performance. Upon the availability of more relevant data, an upgraded version of the ML model can gain more power in terms of predictive accuracy and precision at the patient level and potentially serve as a reliable tool that can be clinically implemented to support precision medicine. A caveat should be given to the in vitro kinase inhibitory dataset used for model building[16]. This dataset includes a mixture of wild-type kinase and mutant kinase that are not strictly disease-associated. There has been recent in vitro profiling of SMKIs against disease-associated mutant kinases[27]. However, most compounds covered in this dataset have not entered clinical trials. The analysis dataset in this report was constructed with the intent to include kinase inhibition data that can maximally cover the approved SMKIs.

The choice of a ML method is contingent on an array of factors including data type, data size, accuracy and precision, and computational efficiency. In addition to the RSF method, other ML techniques including artificial neural network (ANN), support vector machine (SVM), and deep learning methodology can be tuned to conduct time-to-event analyses[28–30]. They are all poised to have a superior predictive performance to the conventional regression-based approaches for "unconventional" survival hazard-predictor relationships, such as ones possessing high nonlinearity. Our simulation-based evaluation has shown comparable performances for RSF and ANN[31]. We also evaluated the data with deep learning (particularly, DeepHit[32] and ANN[28] (please see Supplementary Information section "Evaluation of AE prediction performance using DeepHit and ANN"). The results are similar to ones based on RSF but with much higher computational cost.

ML approaches are inherently non-mechanistic and non-parametric. Consequently, the ML modeling outcomes can be less advantageous than parametric approaches in terms of data interpretation and result extrapolation. This disadvantage of ML approaches could be partially compensated with covariate importance evaluation techniques (e.g., VIMP), the connection weights approach, the partial derivatives for ANN, etc. We adopted the VIMP to identify predictive variables for the interested AEs. For example, hypertension has been recognized as a classic on-target adverse effect as a result of VEGF pathway inhibition[33]. This relationship is consistent with the VIMP analysis results, where VEGFR family kinases including FLT1 (VEGFR1), VEGFR2, and FLT4 (VEGFR) have been identified as the top 1,

2, and 15 kinases associated with hypertension (Fig. 2). Interestingly, the VIMP analysis has also identified an association between JAK2 and hypertension, which is in line with a recent experimental finding that the activation of JAK-STAT pathway plays an important role in the development of Angiotensin II-dependent hypertension[34]. Despite interesting findings, as predictive analysis, the false discovery of KI–AE associations from ML model prediction should be carefully examined, especially given that ML model is a type of method that highly relies on data sufficiency and quality. Conceptually, the determination of false discovery of KI–AE associations requires abundant experimental evidence to demonstrate a certain KI–AE association does not exist. However, it is hard to collect information to determine which identified pair is a false discovery in practice, as most relevant literature reports identified an association between KI and AE, rather than the opposite. As such, our model validation strategy is to evaluate the model-identified KI–AE associations against the ones confirmed in literature reports. The model-predicted KI–AE interaction pairs that have not been experimentally verified can support hypothesis generation for experimentalists and clinicians but still need further validation using clinical/experimental evidence.

Taken together, the developed ML-based KI–AE model can critically serve the scientific community from three fronts. First, the systematically identified KI–AE pairs, as hosted on the interactive application, provide a platform for experimentalists to identify and verify undiscovered KI–AE pairs that can provide critical insight into the origin of drug safety signals. The newly identified KI–AE pairs, yet to be verified, assume reasonable credibility since they are discovered from the same database using the same algorithm and with positive controls of confirmed KI–AE pairs reported in the literature. Second, it can be tuned to a precision-medicine tool to mitigate individual patient safety risks by forecasting clinical safety signals using individual demographic, medication, and drug exposure information. Third, it holds promise as a modern drug development tool to screen and compare SMKI target therapies from a safety perspective.

## Methods

### Data collection and extraction

All FDA-approved SMKIs up to December 31, 2020, were considered. The clinical and non-clinical data of each SMKI were collected. The in vitro kinase inhibitor profiling datasets were curated from the literature and open-access biochemistry databases. Clinical data including PK data, AE records, and patient demographic information were obtained from the clinical trials used to support marketing approval, through FDA internal eCTD (electronic Common Technical Document) software. The study is a meta-analysis across registrational trials, and we did not generate any new clinical data that involves human subjects; therefore no ethical approval was required.

**In vitro kinase inhibitor profiling of SMKIs.** In vitro kinase inhibitor profiling, which assesses the inhibition potency and selectivity of SMKIs against kinase targets, has been developed on a variety of platforms[16,27,35–39]. We selected the dataset[16] that reported the dissociation constants ($K_d$) of SMKIs against 442 kinases using a competitive binding assay because the KI information of the dataset was collected from the same research group and/or experiment and had the maximum coverage of the approved SMKIs. This dataset includes mutated forms of kinases (e.g., EGFR mutations L858R, G719S), which we remained uncombined in our data analysis, considering different mutants may exhibit different clinical therapeutic and safety profiles[40,41]. We also evaluated another dataset[36] that reported the percent inhibition of SMKI against 314 kinases using an enzymatic assay and reported the results in (Supplementary Table 6).

**Definition of AEs.** AE data were retrieved from the safety information reported in the clinical trials of SMKIs. AEs were coded using the Medical Dictionary for Regulatory Activities (MedDRA). AE severity was mainly graded from grade 1 through 5 per the Common Terminology Criteria for Adverse Events (CTCAE, Version 3.0) except for nintedanib with AE severity being described by general categories (i.e., mild, moderate, and severe). Preferred terms coded upon different versions of MedDRA (ranging from 10.0 to 18.1) were normalized and aggregated according to Version 21.0. In addition, a subject expert panel created a customized set of preferred terms to consolidate related AEs, using the similar practice applied for the FDA review of marketing applications to routinely group preferred terms when tabulating AEs rates for the purposes of the risk:benefit analysis as well as for concise labeling on the US Package Insert for oncology drug products. For example, in the review of vandetanib for the treatment of medullary thyroid cancer, Table 26 of the US FDA review notes that "rash" includes the terms "rash, rash erythematous, generalized, macular, maculo-papular, papular, pruritic, exfoliative, dermatitis, dermatitis bullous, generalized erythema, and eczema" (Medical Review of application 022405 available at Drugs@FDA). This grouping of terms is also reflected in the USPI of vandetanib (trade name CAPRELSA, CAPRELSA USPI). We used standard grouping categories employed by the respective review divisions that approved the drug and associated labeling for the matching of AE terms. Other examples of the customized preferred terms include: "vomiting/nausea" encompassing both "vomiting" and "nausea"; "stomatitis" encompassing preferred terms of "mouth ulceration," "oral mucosal eruption," and "stomatitis". Post data processing, we mapped a total of 3996 unique AE terms as reported in the original safety datasets to 2145 customized preferred terms.

**ROR analysis.** We applied ROR analysis to summarize how common an AE was reported for patients taking a particular SMKI compared with the frequency at which the same AE was reported in other SMKIs[42]. A ROR greater than one for a drug–AE combination indicates that the drug of interest had a higher reported frequency of the AE than the rest of the drugs in the comparison group.

### Estimation of individual drug exposure using population PK modeling

We estimated SMKI exposures at the patient level through population PK modeling in consideration of dosing records, demographic information, and other covariate data if applicable. We included all patients who received at least one dose and had at least one evaluable PK measurement. The population PK model was developed using a nonlinear mixed-effect modeling approach implemented in NONMEM (Version 7.4, Icon Development Solutions, Ellicott City, MD). First-order conditional estimation with interaction (FOCEI) was applied in the modeling.

If a population PK study report was included in the New Drug Application (NDA) package, the sponsor-supplied model was reviewed and validated before implementation. Otherwise, self-built population PK models were used to estimate patient-level drug exposures. In this case, a base compartmental model was selected by assessing all conventional model structures in combination via appropriate goodness-of-fit measures as well as visual inspection of diagnostic scatter plots (e.g., observed vs. predicted concentration and residuals/conditional weighted residuals vs. predicted concentration or time plots), parameter estimate precisions, and the minimum objective function values (OFVs). We also accommodated all statistically significant variables that were predictive of individual drug exposures in the final population PK model. Subsequently, individual drug exposures were computed based on the Bayesian post hoc estimates of PK parameters using the final model. In the analysis, we used the computed patient-level average plasma concentration at a steady state ($C_{ave,ss}$) as a measure of individual drug exposure. A total of 16 population PK

models were used for each of the SMKIs included in the final data analyses.

## Modeling safety signals using ML

The workflow of data integration and predictive modeling were illustrated in Fig. 1c. The analyses were conducted using R (Version 3.6.0).

**Data integration.** Clinical and non-clinical data were integrated as the RSF model inputs. Patient demographic information (e.g., age and sex), individual drug exposures, and in vitro KI profiles of the SMKIs were the predictor variables. The empirical measure of KI was computed as $Cave,ssi,j/_{\mathbf{Kdj}}$, where $Cave,ssi,j$ is the average plasma concentration of SMKI $j$ for subject $i$, and $\mathbf{Kd_j} = (Kd_j, Kd_2, \ldots, Kd_m)$ is the vector for the dissociation constants of SMKI $j$ against $m$ kinase targets (the missing values in the in vitro kinase profile is imputed as $K_d = 1e4$), rendering a unique but normalized KI profile for each individual across all SMKIs. As a result, the drug name behind each SMKI becomes irrelevant. Time from treatment start to the first occurrence of an AE event was the response variable (output) of the ML-based survival analysis model. Data censoring occurs when no AE occurred at the end of the study.

**Time-to-event analysis based on RSF.** We implemented RSF[14] as a ML-based approach to model AE occurrence. An RSF was computed by an ensemble of binary decision trees. It can select the most important variables that impact the interested AE events. Bootstrapping and random node splitting were applied to grow an ensemble of independent decision trees to form the RSF. Post the RSF model establishment, variable importance (VIMP)[43] or minimal depth measurement[44] were used to assess the impact of an input variable (i.e., a specific kinase) on the model predictive performance. A large VIMP indicates a strong association between the interested KI and AE; a small or negative VIMP value indicates low or no association, while a small minimal depth indicates a strong association. We also applied a recently developed deep-learning-based survival analysis method (i.e., DeepHit) and ANN to compare prediction performance against RSF.

**Performance evaluation of the predictive survival model.** Concordance index (C-index) was used to assess the performance of the predictive survival model. C-index is related to the area under the receiving operating characteristic (ROC) curve and is commonly used in prediction error estimation. C-index theoretically ranges between 0 and 1 with the value of 0.5 representing the performance of a random model and 1 representing a perfect prediction. In addition, C-index does not depend on a single fixed time for evaluation and accounts for the presence of censoring.

Two evaluation approaches were used to examine the predictive performance of an established model. To evaluate patient-level AE predictive performance, we adopted the leave-one-out (LOO) cross-validation approach, where one of the patient data was left out as the testing data, and the rest of the data were used as training data to establish the ML survival model. Each patient data was rotated as the testing data so as to conduct predictions for all patients based on the model developed from the corresponding training datasets. To evaluate AE predictive performance on the patient population level, a bootstrapping approach was applied with 500 iterations. In each iteration, the original data were split into a bootstrap training dataset and a corresponding testing dataset. Samples of bootstrap training dataset were drawn without replacement from the original data. We first trained the predictive model with the bootstrap training dataset, then predicted the outcomes of the testing dataset. The predictive performance was subsequently measured by the C-index. The bootstrapping estimate of the prediction accuracy was calculated by averaging the values of C-index from all the iterations.

**External model validation on independent datasets.** In addition to the cross-validation, we undertook an external validation on clinical datasets of two different SMKIs, neratinib and imatinib, apart from the 16 SMKIs used for training the predictive ML model. These two SMKIs are selected for the analysis because they have accessible KI information ($K_d$), as well as patient-level exposure and safety data available for certain clinical trials. Patient-level drug exposure data were not collected for the pivotal studies but were available for other Phase II/III monotherapy studies for neratinib and imatinib[45–47]. We applied the previously trained ML model to predict AE probabilities for patients in these studies, identified the AEs with high probability, and compared them with the most frequently reported (reported for 10% of subjects) TEAEs in the corresponding study report as an assessment of model performance.

## Reporting summary

Further information on research design is available in the Nature Research Reporting Summary linked to this article.

## Data availability

The data used in this study were derived from the third-party registrational clinical trials data (including individual-level data), which cannot be made publicly available due to confidentiality obligations and regulation policies in FDA. The processed datasets may be available under restricted access for bona fide research. Specific requests for de-identified population-level data should be sent to the corresponding author upon ethical review. All requests will be promptly reviewed within 15 working days. We have made secondary data and information publicly accessible via the web application "Identification of Kinase-Specific Signal" (https://gongj.shinyapps.io/ml4ki).

## Code availability

The custom codes used in this manuscript are available on the GitHub page https://github.com/xj-gong/ml4ki.

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

## Acknowledgements

This work was supported by the FDA Critical Path Initiative awarded to L.Z. The authors are thankful to Jason Jiang, Yizhou Wang, and Edward Kim who provided assistance in the research and to Mingjiang Xu for insightful comments on the manuscript.

## Author contributions

X.G., M.H., J.L., G.K., A.M., and L.Z. conceptualized the study. X.G., M.H., J.L., G.K., and L.Z. collected and analyzed the data. J.X., T.P., and A.d.C. contributed to the interpretation of the results and writing the manuscript. X.G., M.H., and L.Z. took the lead in writing the first draft. X.G., M.H., J.L., G.K., and L.Z. had full access to all the data in the study. All authors critically reviewed, edited, and approved the final manuscript.

## Competing interests

G.K., J.X., T.P., A.M., and T.P. contributed to the work in their previous employment with the Food and Drug Administration. They do not have access to data and have no major involvement in the work after leaving the agency. Currently, G.K. and T.P. are employed with Beigene with possession of shares; J.X. is the clinic owner of Potomac Oncology and Hematology; A.M. is employed with Paraxel with possession of shares.
