## [Peer Review File · Nature Communications]

Reviewers' Comments:

Reviewer #1:

Remarks to the Author:

The manuscript with the title "Big (Registrational) Data based Decoding of Kinase-Adverse Event Associations for Small Molecule Kinase Inhibitors" integrated a multi-domain dataset and proposed a computational model for predicting the relationships between kinase targets and adverse events (AEs). The authors also provided a simple platform to identify undiscovered KI-AE pairs. However, the manuscript still has drawbacks and is below the acceptance threshold. My major concerns are listed as follows:

1. The proposed model seems to just apply some existing algorithms such as RSF to this field, which lacks innovation. As a tool, the authors should explain in detail why it can achieve good results from the principle or mechanism.
2. Although the authors mentioned that other ML techniques can be tuned to conduct time-to-event analyses, they did not compare RSF with other methods under the same experimental setting. Therefore, it seems that the experiment is not enough to support the conclusion.
3. In the section Method, the proportional reporting ratio (PRR) does not estimate relative risk. The reporting odds ratio (ROR) should be analyzed and discussed.
4. The authors employed 10% incidence as a threshold to verify the performance of the model in this manuscript. How is the model performance under different thresholds?
5. In the treatment-emergent AE vomiting occurrence prediction module of the platform, some problems need to be solved: (1) the corresponding kinase inhibitor and gender are missing in the table name of the occurrence prediction chart. (2) the download function of the prediction results needs to be provided. (3) The chart name of prediction results should be updated after clicking the 'Predict' button.

Reviewer #2:

Remarks to the Author:

In this manuscript, Gong et al attempted to establish kinase-adverse event (KI-AE) associations for small molecular kinase inhibitors from clinical studies. Here, the authors focused on 65 small molecule kinase inhibitors (SMKIs) approved by the FDA, and using 16 of the SMKIs registrational studies to extract patient-level PK and AE data (n = 4638 patients). The in vitro KI profiling data for these SMKIs were obtained from previously published paper. The authors claimed that they normalized the AEs to MedDRA terms version 21.0, and the grading of AEs according to CTCAE, and obtained 2145 preferred AE terms. The authors used a non-linear mixed-effect modeling approach implemented in NONMEM to estimate individual drug exposure using population PK Modeling. Specifically, First-order conditional estimation with interaction was applied here. The authors then used a Random Survival Forest (RSF) model to integrate clinical and non-clinical (drug exposures, in vitro KI profiles) features. Overall, the research is interesting and addressing a critical clinical problem for drug development. However, the current manuscript suffers several serious weaknesses. In addition, no new KI-AEs were provided.

Major comments:

1. It is unclear how the authors handled missing values in the in vitro Kinase profiles. It is also unclear in Suppl Table 1, how do the authors determined the main targets of the SMKIs, is it based on certain cut-off of the Kd values?
2. the kinase profiles have some mutated forms of kinases, and it seems like the authors are treating those as individual variables (Figure 2 a), what is the rationale of doing that is not clearly defined.
3. No comparisons with some other methods proposed previously to correlate KI-AE, for example (PMID: 20434586, PMID: 27631620).
4. From the KI-AE interaction pairs, how do the authors account for false discovery?

5. The authors used some examples to highlight / validate the prediction, however, the authors should perform systematic comparison with AEs reported in FAERS or SIDER database.

6. How do the authors performed the AEs matching is not clearly described, the authors claimed that they are using MedDRA as a reference, and more details are needed.

7. The data used in the study is not publicly available.

Reviewer #3:

Remarks to the Author:

The manuscript by Gong et al., describes the development of a machine learning model to identify associations between reported adverse events (AEs) in clinical studies with inhibition of specific kinases. They leverage existing data sets in which the kinase targets of clinical kinase inhibitors have been evaluated on a large scale. They validate their model by recapitulating a number of previously reported associations and develop a web-based tool to allow free access to the data. The use of a pharmacokinetic (PK) model to determine drug exposure as part of the analysis is an innovative and novel component. The PK model allows integration with target inhibition EC50 values from pan-kinome drug profiling studies. The methodology is clearly described, the source code was provided to reviewers but should also be made available through a direct link in the manuscript. Overall, the approach and data are potentially of value to clinical care of patients treated with small molecule kinase inhibitors and also will facilitate the discovery of new kinase-associated toxicities.

My primary concern is that the manuscript would benefit from a deeper analysis of the data. To demonstrate its utility, for example, the authors should move beyond validation of known associations and report (and discuss) one or two novel kinase-AE associations they have uncovered in the data. Clearly the scope of the data is too large for a comprehensive analysis but some new kinase-AE prediction(s) would be welcome to illustrate the value of the data. Overall, I expect this to be a valuable resource for clinicians/investigators interested in exploring their own drugs/kinases/AEs of interest.

Additional points

In Figure 1b the authors compare AE proportional reporting ratios for 4 EGFR inhibitors and report that they share similar incidences of diarrhea, fatigue, and vomiting, which are known "on target" effects of EGFR inhibition. Interestingly, these symptoms exhibit only moderate reporting ratios. How do the authors think about/prioritize "similarity" in PRR score versus higher actual scores? What about PPR scores that are "similar" but very low? Why do they not discuss other AEs with similar scores such as cough or decreased appetite?

Using their machine learning model, the authors claim to have identified "all the pairs [of kinases and SMKIs] that have been experimentally proved". To support this claim they cite a 2013 study (reference 6) although their own data addresses kinase inhibitors/AEs reported since 2020. Certainly, additional kinase-AE associations have been described since 2013. While the criteria the authors used for "experimentally proved" are not described, it seems like this is a missed opportunity to investigate other reported kinase-associated toxicities. A more comprehensive analysis of associations described in the recent literature is required along with an analysis of whether they were identified by the machine learning analysis. If not, how do the authors explain the discrepancy?

The utility of Time to AE onset data in Table 1 is unclear to me, given that the range of values is so large (typically 1 day to ~ 2 years!). A more granular presentation of the data (for example a histogram of values), perhaps in supplemental data, would be welcome to show the distribution of the data.

In Figure 3, while model prediction of TEAEs was generally very good, it would be worth a deeper analysis of specific TEAEs that were not well predicted by the model. In particular, can the authors speculate why certain TEAEs with high observed frequency were not well predicted by the model?

The web application the authors report is simple and user friendly. I can imagine some additional ways of interrogating the data that are not permitted in the current version that could be considered. For example, currently the data can be searched for a specific AE (with the results returning a ranked list of associated kinases). It could be valuable to allow the reverse search, in which an investigator searches for a kinase of interest and the software would return a ranked list of associated AEs.

Title: Big (Registrational) Data based Decoding of Kinase-Adverse Event Associations for Small Molecule Kinase Inhibitors

We would like to express our appreciation to the editors and reviewers for their constructive comments, which have critically guided us to further improve the quality of the manuscript. We have carefully revised the manuscript and included new insights and further data analyses.

Our point-by-point responses to each of the reviewers' comments are listed below. Figures and tables for this response letter are listed at the end of the response, and are numbered as **Figure L1**, 2, etc., and **Table L1**, 2, etc.

The revised/added text **is indicated in tracked changes in the manuscript.**

Reviewer #1 (Remarks to the Author):

The manuscript with the title “Big (Registrational) Data based Decoding of Kinase-Adverse Event Associations for Small Molecule Kinase Inhibitors” integrated a multi-domain dataset and proposed a computational model for predicting the relationships between kinase targets and adverse events (AEs). The authors also provided a simple platform to identify undiscovered KI-AE pairs. However, the manuscript still has drawbacks and is below the acceptance threshold. My major concerns are listed as follows:

1. The proposed model seems to just apply some existing algorithms such as RSF to this field, which lacks innovation. As a tool, the authors should explain in detail why it can achieve good results from the principle or mechanism.

Response:

We appreciate the comment and agree that further work and discussion should be done for other ML methods. Following the reviewer's comment, we have applied a recently developed deep learning survival analysis method, DeepHit, as an alternative ML method for data analysis. Our results showed that performances of RSF and DeepHit are comparable (see response to Question 2 for details). Of note, DeepHit is more computationally expensive than the RSF approach, as indicated by a ~20X longer running time. The overall finding is consistent with findings from one of our earlier publications¹. In the referenced effort, we conducted a performance check across survival analysis methods by different algorithms (e.g., RSF, ANN and Cox models) and found that RSF outperforms Cox and ANN models for nonlinear and/or high-dimensional data. Overall, we consider that RSF is a non-inferior option to other ML approaches. To address the comments, we added the details of the above analyses in the **Supplementary Information** section “Evaluation of AE prediction performance using DeepHit and ANN”, and revised the relevant paragraph in the *Discussion* section **(please see lines 240-243).**

2. Although the authors mentioned that other ML techniques can be tuned to conduct time-to-event analyses, they did not compare RSF with other methods under the same experimental setting. Therefore, it seems that the experiment is not enough to support the conclusion.

Response:

As mentioned in response to the first comment, we have applied a recently developed deep-learning based survival analysis method (i.e., DeepHit) to our data and compared performances between RSF and DeepHit. We also added to the performance evaluation a well-developed ML method, artificial neural network (ANN), which had been used for ML method demonstration for time-to-event analysis in our previous simulation work¹.

The new analysis outcomes have been consolidated to the **Supplementary Information** “Evaluation of AE prediction performance using DeepHit and ANN” and been referenced in the *Discussion* section (lines 240-243). For reviewing convenience, the content is provided below as well.

DeepHit

Unlike conventional survival models that rely on strong parametric assumption, DeepHit uses a deep neural network to learn the distribution of survival times directly and makes no assumptions about the underlying stochastic process and allows for the possibility that the relationship between covariates and risk(s) changes over time. Like parametric approaches, DeepHit can accommodate multiple competing risks².

An exemplary architecture of a DeepHit model dealing with two events of interest is shown in Figure L1. The DeepHit model consists of multiple fully connected subnetworks comprising a shared sub-network and multiple cause-specific sub-networks. The shared sub-network inputs information from covariates (X), and the cause-specific sub-network outputs information from the shared sub-network and covariates to the outcome information of the specific event. Specifically, to handle survival data, the DeepHit model takes in a covariate matrix X, and outputs a vector y where each element $y(k, t)$ represents the probability of a certain event k (e.g., death or cured) occurring at time t. Covariate matrix X first enters one shared subnetwork, and then gets through cause-specific subnetworks for the competing risks in evaluation. To make sure information in the original covariate matrix X is not lost, there is a residual connection between X and each of the cause-specific subnetworks. The output layer is a SoftMax layer, so the sum of $y(k, t)$ equals 1. The DeepHit model (<https://github.com/chl8856/DeepHit>) has been described in detail previously³.

The DeepHit model was applied to our data for performance comparison. We customized the DeepHit code shared by the developer and conducted data analysis in a TensorFlow environment. For hyper-parameter finetuning, we adopted a random search strategy, consisting of 20 random search iterations. A random combination of the following hyperparameters is selected in each iteration: # of neurons and # of layers in shared and cause-specific subnetworks, activation function, and beta (i.e., a value of the ranking loss function that is adapted from the idea of concordance and represents number of correctly ordered pairs). After building a model for each of the 20 random combinations of the hyperparameters, the models are evaluated on the test dataset, and the hyperparameters corresponding to the model with the highest C-index on the testing set are selected. We built one DeepHit model for each adverse event with its own set of hyperparameters.

ANN

ANNs mimic brain networks to implement artificial intelligence learning. An ANN typically consists of three layers: input (containing nodes for the predictor variables), hidden (containing nodes for mapping input to output layers), and output (containing nodes for the prediction outcomes). The nodes between adjacent layers are fully connected. The strength of each connection is usually represented by an adjustable weight parameter. Each node in the hidden and output layers can serve as a nonlinear activation function (e.g., logistic function), of the weighted linear combination of inputs from the previous layer. Application of the standard ANN to survival data is not straightforward and requires adaptation. In our study, we used the partial logistic regression approach ANN (PLANN)⁴ based on a three-layer, feed-forward neural network among previously proposed ANN strategies for survival analysis⁵. Briefly, to accommodate survival data in ANN, the input layer in this model includes both the predictor variables of interest and the time variable. A series of non-overlapped time intervals covering the duration of the study are predefined. For each subject, the values of his/her predictor variables are replicated for all time intervals. Uncensored cases are repeated until a time interval is reached in which the event is observed. During training, the output layer uses '0' for each time interval without event, and '1' for the time interval with event. Given a trained PLANN, the output can yield the approximated hazard function and survival function. In principle, PLANN model allows the joint modeling of time and the continuous and categorical covariates as input predictors without proportionality constraints.

Results

We evaluated the predictive performance of the ML-based methods using the same dataset described in Table L1. For each AE, 80% of the dataset were bootstrap-sampled as the training dataset, and the remaining 20% of the dataset were used for model validation. Table L1 shows the predictive performance as measured by C-index and 90% confidence interval. In addition to the five representative AEs in the main text, we investigated five more AEs to compare the prediction performances across different ML methods. The three ML methods showed generally similar performances. While no ML method stands out as a clear "winner" in terms of prediction performance, RSF seems to perform slightly better by providing more highest C-index scores among all the cases (i.e., 4 out of 10) compared to other two methods (i.e., 3 out of 10).

Caveat should be given to the use of DeepHit. As a method not originally developed for the purpose, it can be challenging to use DeepHit to determine the variable importance. To counter the challenge, Dynamic DeepHit, an extension of DeepHit, was developed to assess variable importance². Briefly, to test the importance of a variable, two models are established by using the maximum and minimum values of the interested variables individually. The variable importance is represented by calculating the difference in cumulative incidence functions of the two models. However, we found that the variable importance algorithm for Dynamic DeepHit can only effectively identify important variables when these variables are linearly related to dependent variables. In summary, an efficient variable importance method for DeepHit has not been developed. The permutation approach could be another option for variable importance analysis, but further efforts are warranted.

3. In the section Method, the proportional reporting ratio (PRR) does not estimate relative risk. The reporting odds ratio (ROR) should be analyzed and discussed.

Response:

We appreciate the reviewer's comment to add ROR analysis and have applied ROR to the dataset used for Figure 1b in the manuscript. ROR and PRR generated similar results in this case (Figure L2). We revised relevant paragraph in the *Result* section (line 80-81) and *Method* section (line 315-319) and updated Figure 1b in the manuscript with results based on ROR.

Specifically, for each AE, we further calculated the coefficient of variation (%CV) of ROR for vandetinib, afatinib, erlotinib and nintedanib, as a measure of heterogeneity for the risk of AE occurring across these four SMKIs targeting EGFR. The AEs marked bold in Figure L2 (i.e., "decreased appetite", "vomiting", "diarrhea", "fatigue" and "nausea") are associated with the lowest five %CV values, indicating the four EGFR inhibitors shared similar relative risks of these on-target AEs. Other AEs exhibit higher %CVs, with nine AEs having %CVs larger than 100%, suggesting varying off-target effects for the four EGFR inhibitors.

4. The authors employed 10% incidence as a threshold to verify the performance of the model in this manuscript. How is the model performance under different thresholds?

Response:

We employed 10% incidence as a threshold in the external validation because the reported incidence in $\geq 10\%$ subjects was used as criterion for "frequently-reported treatment emerged adverse event (TEAE)" in the clinical study reports of neratinib and imatinib. Following the reviewer's comment, we conducted further analyses using two additional threshold values (i.e., 15% and 20%). Consistent results were observed under all three threshold percentages used for analysis. The details are described as follows.

Figure L3 shows the results of model performance under different thresholds. When the threshold increases from 10% to 15% and 20%, the model predicted AEs with high probabilities can still cover all the frequently reported TEAEs in neratinib studies with only one exception TEAE frequently reported in the first neratinib study (Panel a), indicating good sensitivity of the model to the observation threshold. For the imatinib study, the relationship between the predicted AEs with high probabilities and frequently reported TEAEs becomes statistically insignificant with increasing thresholds (Panel c). For all the selected threshold values, although there are AEs with model predicted probability above and observed incidence below threshold values, the model specificities (measuring the proportion of true negatives that are correctly identified by the model) are all above 95%, considering thousands of AEs are true negatives with predicted probabilities below the threshold. The model predicted AE probabilities remain significantly correlated with the observed values for the two neratinib studies. Overall, the model specificity is not compromised against varying threshold values.

The above results were added as Supplemental Figure S3, and the manuscript was updated accordingly (see line 164-165).

5. In the treatment-emergent AE vomiting occurrence prediction module of the platform, some problems need to be solved: (1) the corresponding kinase inhibitor and gender are missing in the

table name of the occurrence prediction chart. (2) the download function of the prediction results needs to be provided. (3) The chart name of prediction results should be updated after clicking the 'Predict' button.

Response:

Following the reviewer's comments, we have updated the "Treatment-emergent AE prediction" module of <https://gongj.shinyapps.io/ml4ki/>. The screenshot below shows an example: (i) the name of the occurrence prediction chart is updated after clicking "Predict" button; (ii) the name now includes the AE (e.g., nausea), patient information (e.g., female, 37yo) and the administered kinase inhibitor (e.g., bosutinib); (iii) download function is provided to allow saving the chart view into PNG or JPEG, or saving its data in CSV; annotation function is also provided.

Reviewer #2 (Remarks to the Author):

In this manuscript, Gong et al attempted to establish kinase-adverse event (KI-AE) associations for small molecular kinase inhibitors from clinical studies. Here, the authors focused on 65 small molecule kinase inhibitors (SMKIs) approved by the FDA, and using 16 of the SMKIs registrational studies to extract patient-level PK and AE data (n = 4638 patients). The in vitro KI profiling data for these SMKIs were obtained from previously published paper. The authors claimed that they normalized the AEs to MedDRA terms version 21.0, and the grading of AEs according to CTCAE, and obtained 2145 preferred AE terms. The authors used a non-linear mixed-effect modeling approach implemented in NONMEM to estimate individual drug exposure using population PK Modeling. Specifically, First-order conditional estimation with interaction was applied here. The authors then used a Random Survival Forest (RSF) model to integrate clinical and non-clinical (drug exposures, in vitro KI profiles) features. Overall, the research is interesting and addressing a critical clinical problem for drug development. However, the current manuscript suffers several serious weaknesses. In addition, no new KI-AEs were provided.

Major comments:

1. It is unclear how the authors handled missing values in the in vitro Kinase profiles. It is also unclear in Suppl Table 1, how do the authors determined the main targets of the SMKIs, is it based on certain cut-off of the Kd values?

Response:

We imputed $K_d=1e4$ for the missing values in the in vitro kinase profile, which has now been explicitly specified in the *Method* section (see lines 350-351).

The main targets of the SMKIs were determined using information in the drug labels as reference. For instance, in axitinib tablets drug label (https://www.accessdata.fda.gov/drugsatfda_docs/label/2012/202324lbl.pdf), the mechanism of action in “Section 12 Clinical Pharmacology” states that “*Axitinib has been shown to inhibit receptor tyrosine kinases including vascular endothelial growth factor receptors (VEGFR)-1, VEGFR-2, and VEGFR-3 at therapeutic plasma concentrations.*” Therefore, in Supplemental Table S1 we listed “VEGFR” as the main target for axitinib. We clarified in the footnote for Supplemental Table S1.

2. the kinase profiles have some mutated forms of kinases, and it seems like the authors are treating those as individual variables (Figure 2a), what is the rationale of doing that is not clearly defined.

Response:

In the manuscript, we selected the in vitro kinase inhibitor profiling dataset⁶ that reported the dissociation constants (Kd) of 442 mutated forms of kinases, because its kinase inhibition information was collected from the same research group and/or experiment and had the maximum coverage of the approved SMKIs. Mutated forms of kinases may exhibit different clinical therapeutic and safety profiles.

Using EGFR and its mutated forms as an example, we found that:

1. There is an unmet clinical need to classify EGFR mutants by understanding their drug selectivity⁷.
2. The EGFR mutations mentioned in our manuscript either reside adjacent to or are part of the binding pocket of ATP-competitive inhibitors^{8, 9}.
3. The activity of the EGFR downstream signaling pathway is largely dependent on which mutation the receptor harbors.
 - The EGFR mutations have different drug selectivity. For instance, gefitinib and erlotinib (first-generation EGFR-TKIs) were particularly beneficial in patients with sensitizing mutations (L848R and the exon-19 deletion)^{10, 11}, while the mutation T790M acquired resistance to these inhibitors¹².
 - The EGFR mutations are associated with the severity of AEs. For example, L858R destabilizes the ligand binding domain, causing the constitutive activity of the receptor⁹ and potentially changing the severity of its coupled AEs. It will be helpful for clinicians to understand the severity of the AEs associated with the mutations. Our method is intended to serve this purpose (Figure 2 in the main text).
 - The EGFR mutations potentially have different AE profiles. Different mutants have distinct activating mechanisms⁹, which can lead to different AEs.

Therefore, we considered it reasonable not to combine mutated forms of kinases⁹. The manuscript has been updated to better clarify this point (see lines 288-291).

3. No comparisons with some other methods proposed previously to correlate KI-AE, for example (PMID: 20434586, PMID: 27631620).

Response:

We acknowledge that the reviewer's mentioned other methods to correlate kinase inhibition and adverse events. We added relevant text regarding the comparisons in the *Discussion* section (see lines 205-214).

In particular, Yang et al. (PMID: 20434586)¹³ identified associated AE and kinase targets (AE-KT) pairs by extracting (i) a matrix of AE frequency for SMKIs from literature mining and (ii) a matrix of SMKIs inhibition for KT (in terms of Kd) from published in vitro assay, and multiplying the two matrices to obtain an AE-KT association score matrix. The higher the score indicates the more prioritized the association of a KT with the corresponding AE.

In our previous work^{14, 15}, we established a matrix of AE-kinase pairs in a similar fashion to Yang et al., 2010 and applied disproportionality analyses to identify significant KI-AE associations (<https://jzliu.shinyapps.io/KINASE/>). Our current study shares similarity with publication¹³ by integrating published in vitro kinome data (Kd) although with increased scale (assay of 442 kinase targets⁶ used in our manuscript vs. 266 targets¹⁶ used in Yang's paper). Besides, Yang and colleagues considered AEs curated from literatures at the population level, and their research was limited by access to patient level clinical data. Our current study extracted clinical response from patient level clinical trial data, which not only expanded the findings to 2,145 AEs (compared to 71 AEs in paper¹³) but also provided a framework to support the individualized prediction of AE based on the KI profiling and drug exposure level.

Federer et al. (PMID: 27631620)¹⁷ focuses on the identification of the drug-AE pairs rather than the kinase-AE pairs as we are evaluating in our study. As such, their work does not offer a framework to support AE prediction at individual subject level.

4. From the KI-AE interaction pairs, how do the authors account for false discovery?

Response:

Determination of false discovery requires abundant clinical evidence to demonstrate a certain KI-AE interaction does not exist. It is hard to collect information to determine which identified pair is a false discovery, as the majority of relevant literature reports identified association between KI and AE, rather than the opposite. As such, our model validation strategy is to evaluate the model-identified KI-AE associations against the ones confirmed in literature reports, as shown in Figure. 2. Furthermore, in the point-by-point response to one of Reviewer 3's comments ("*... A more comprehensive analysis of associations described in the recent literature is required along with an analysis of whether they were identified by the machine learning analysis...*"), we conducted a comprehensive survey on the publications after 2013 for further model validation. As shown in our response to the comment, the updated results demonstrate that the developed ML model still can effectively identify those KI-AE interactions.

Given that ML model is a type of method that highly relies on data sufficiency and quality, false discovery from ML model prediction cannot be completely ruled out. As stated, the model-predicted KI-AE interaction pairs that have not been experimentally verified can support hypothesis generation for experimentalists and clinicians but still need further validation using clinical/experimental evidence.

We added relevant text regarding the false discovery in the *Discussion* section (see lines 255-265).

5. The authors used some examples to highlight / validate the prediction, however, the authors should perform systematic comparison with AEs reported in FAERS or SIDER database.

Response:

We thank the reviewer for the valuable comments on comparing the predictions from ML models with AEs reported from other existing databases like FAERS or SIDER. We performed the comparison as follows:

We first conducted queries in the U.S. FAERS database for the safety profile regarding the two SMKIs neratinib and imatinib. The search parameters are listed In Table L2. Based on the query results, we extracted the total number and percentage of the reported AEs.

To compare the AEs reported in FAERS with our predictions, we used the same data used to generate Figure 3 in the manuscript, specifically data from patients suffering from:

- breast cancer in two neratinib monotherapy studies (116 and 136 patients respectively),
- chronic myelogenous leukemia in an imatinib monotherapy study (349 patients).

We applied the trained ML model to predict patient-level AE probabilities for these studies and then calculated the averaged AE prediction across study population.

Of note, the ML predictive model in this study was based on randomized, controlled clinical trial data while the FAERS data is derived from a spontaneous post-marketing reporting system. Given the different nature of the data, we did not directly compare their results by absolute percentages but conducted comparative analysis by calculating (i) hit rate of ML model predicted AEs with high probability ($\geq 10\%$) in a top 20 reported AEs in FAERS (two columns from the rightmost in Table L3), (ii) Spearman rank correlation coefficient (Spearman ρ) of the occurrence percentages between the ML model predicted AEs and the top 20 reported AEs in FAERS. Spearman ρ is used as it is based on rank order of the data, and can accommodate a certain level of nonlinear correlation, which is considered more appropriate for real-world situation.

Table L3 shows results of whether an AE with ML model prediction of high probability (i.e., $\geq 10\%$) appears in the top 20 reported AEs in FAERS (“Yes” in 1st column). For neratinib study I (Table L3.a), among the top 20 reported AEs in FAERS, 13 AEs were predicted as AEs with high probability ($\geq 10\%$), 4 AEs were not reported in the clinical studies. Though there are three mismatched AEs (i.e., dehydration, abdominal discomfort, and muscle spasms), they are at the relatively lower ranks in the FAERS top 20 list (i.e., 6.6%, 4.8% and 4.2%). Similar results were observed for neratinib study II (Table L3.a) and imatinib study (Table L3.b).

To facilitate the correlation analysis, we generated scatter plots of reported percentages of top AEs in FAERS vs. ML model predicted probabilities of these AEs. The Spearman ρ was used to quantify the correlation by accommodating the potential nonlinear relationships (Figure L4). The results for all three studies indicate a clear correlation pattern (not purely linear) with significant correlations ($p < 0.05$).

Overall, the majority of top FAERS-reported AEs were captured by ML model with predicted high probability. Notably, for AEs specific to pharmacovigilance or AE with preferred terms not reported in clinical trials of the 16 SMKIs used for training the ML model, a predictive ML model cannot be built, and therefore no prediction can be made for these AEs.

We added the details of the above analyses in the **Supplementary Information** section “Comparison of the predictions from ML models with AEs reported in FAERS” and revised the relevant paragraph in the *Results* section (please see lines 167-174).

6. How do the authors performed the AEs matching is not clearly described, the authors claimed that they are using MedDRA as a reference, and more details are needed.

Response:

The Medical Dictionary for Regulatory Activities (MedDRA) is the controlled terminology standard required by the FDA for the coding of AEs for Investigational New Drug Applications, New Drug Applications, or Biologic Licensing Applications. This standard permits the aggregation and analysis of AEs across applications and trials even if the data were submitted by different sponsors.

The MedDRA hierarchy is comprised of five levels, ranging from very specific to very general:

1. “Lowest Level Terms” (LLTs, most specific level): more than 80,000 terms, reflecting how an observation might be reported in practice. This level directly supports assigning MedDRA terms within a user database.

2. "Preferred Terms" (PTs) are distinct descriptors (single medical concept) for a symptom, sign, disease diagnosis, therapeutic indication, investigation, surgical or medical procedure, and medical social or family history characteristic. Each LLT is linked to only one PT. Each PT has at least one LLT (itself) as well as synonyms and lexical variants (e.g., abbreviations, different word order).
3. "High Level Terms" (HLTs) groups related PTs based upon anatomy, pathology, physiology, aetiology or function.
4. "High Level Group Terms" (HLGTs) are related HLTs.
5. "System Organ Classes" (SOCs) groups HLGTs by aetiology (e.g. *Infections and infestations*), manifestation site (e.g. *Gastrointestinal disorders*) or purpose (e.g. *Surgical and medical procedures*). SOC to contain issues pertaining to products and one to contain social circumstances."

We used the MedDRA "Preferred Term" for the AE analysis; however, as described in the manuscript, certain terms such as 'stomatitis' may encompass related AEs. It is standard practice for the FDA review of marketing applications to routinely group preferred terms when tabulating AEs rates for the purposes of the risk:benefit analysis as well as for concise labeling on the US Package Insert for oncology drug products. For example, in the review of vandetanib for the treatment of medullary thyroid cancer, Table 26 of the US FDA review notes that rash includes the terms "rash, rash erythematous, generalized, macular, maculo-papular, papular, pruritic, exfoliative, dermatitis, dermatitis bullous, generalized erythema and eczema" (Medical Review of application 022405 available at Drugs@FDA). This grouping of terms is also reflected in the USPI of vandetanib (trade name CAPRELSA, CAPRELSA USPI). We used standard grouping categories employed by the respective review divisions that approved the drug and associated labeling for the matching of AE terms.

For clarity, we incorporated the details on AE preferred terms mapping into the *Method* section (see lines 302-313).

7. The data used in the study is not publicly available.

Response:

The data used for this manuscript were derived from the third-party registrational clinical trials data that include proprietary and business sensitive information from pharmaceutical companies, and thus cannot be made publicly available. The data will be made available if requested by the editors and reviewers, within the terms of data use agreement and if compliant with ethical and legal requirements. Notably, secondary data and information is publicly accessible in the platform (<https://gongji.shinyapps.io/ml4ki>), as reported in the manuscript.

Reviewer #3 (Remarks to the Author):

The manuscript by Gong et al., describes the development of a machine learning model to identify associations between reported adverse events (AEs) in clinical studies with inhibition of specific kinases. They leverage existing data sets in which the kinase targets of clinical kinase inhibitors have been evaluated on a large scale. They validate their model by recapitulating a number of previously reported associations and develop a web-based tool to allow free access to the data. The use of a pharmacokinetic (PK) model to determine drug exposure as part of the analysis is an innovative and novel component. The PK model allows integration with target inhibition EC50 values from pan-kinome drug profiling studies. The methodology is clearly described, the source code was provided to reviewers but should also be made available through a direct link in the manuscript. Overall, the approach and data are potentially of value to clinical care of patients treated with small molecule kinase inhibitors and also will facilitate the discovery of new kinase-associated toxicities.

My primary concern is that the manuscript would benefit from a deeper analysis of the data. To demonstrate its utility, for example, the authors should move beyond validation of known associations and report (and discuss) one or two novel kinase-AE associations they have uncovered in the data. Clearly the scope of the data is too large for a comprehensive analysis but some new kinase-AE prediction(s) would be welcome to illustrate the value of the data. Overall, I expect this to be a valuable resource for clinicians/investigators interested in exploring their own drugs/kinases/AEs of interest.

Response:

We thank the reviewer's constructive comment. To address the comment, we have conducted additional analyses in this revision including (1) comparing the predictions from ML models with AEs reported from FAERS (please refer to the response to question 5 from the reviewer #2), and (2) conducting a new literature survey and comparing the newly verified KI-AE pairs from the survey with the ML identified pairs not listed as experimentally verified in the last draft (please refer to our response below).

Additional points

In Figure 1b the authors compare AE proportional reporting ratios for 4 EGFR inhibitors and report that they share similar incidences of diarrhea, fatigue, and vomiting, which are known "on target" effects of EGFR inhibition. Interestingly, these symptoms exhibit only moderate reporting ratios. How do the authors think about/prioritize "similarity" in PRR score versus higher actual scores? What about PPR scores that are "similar" but very low? Why do they not discuss other AEs with similar scores such as cough or decreased appetite?

Response:

We appreciate the reviewer's observation and comments. In Figure 1b in the main text, PRR score are replaced by ROR score following the notion that the latter evaluates relative risk. We calculated the coefficient of variations (%CVs) of ROR scores for vandetinib, afatinib, erlotinib

and nintedanib, as a measure of similarity for the risks of AEs occurring across these four SMKIs targeting EGFR. Along with the AEs “vomiting”, “diarrhea”, “fatigue” originally reported in the manuscripts, “decreased appetite” and “nausea” were found to be associated with the lowest five %CV values, indicating the four EGFR inhibitors shared similar risk of on-target AEs. Other AEs are associated with higher %CVs, with nine AEs having %CVs of more than 100%, suggesting varying off-target effects for the four EGFR inhibitors. We edited the *Results* section **line 82** to add the two on-target AEs that share similar risk across four inhibitors but were not reported in the original manuscript.

Using their machine learning model, the authors claim to have identified “all the pairs [of kinases and SMKIs] that have been experimentally proved”. To support this claim they cite a 2013 study (reference 6) although their own data addresses kinase inhibitors/AEs reported since 2020. Certainly, additional kinase-AE associations have been described since 2013. While the criteria the authors used for “experimentally proved” are not described, it seems like this is a missed opportunity to investigate other reported kinase-associated toxicities. A more comprehensive analysis of associations described in the recent literature is required along with an analysis of whether they were identified by the machine learning analysis. If not, how do the authors explain the discrepancy?

Response:

We appreciate the reviewer’s critique and comment of using recently reported AE-Kinase association to validate the developed ML model.

In the original manuscript, we used “experimentally proved” to describe those KI-AE associations that are well-established through clinical studies and practice (e.g., VEGFR-hypertension, EGFR-diarrhea), referencing the 2013 review article¹⁸ which summarizes kinase targets and associated toxicities. For clarity, we revised the *Results* section **line 95-96** to “... covered all pairs that have been experimentally well-established ones [6], including VEGFR-hypertension ...”.

Following the reviewer’s comment, a comprehensive literature survey has been conducted as follows:

1. We conducted a query of PubMed database with Boolean formula (((adverse event[OT]) OR (adverse effect[OT]) OR (adverse reaction[OT]) OR (toxicities[OT])) AND (kinase [OT]) AND 2013:2022 [dp]) and collected 88 papers as a result
2. Out of the 88 papers, 63 papers describing association between kinase and AEs in their abstracts were selected for further review.
3. In addition, 15 papers citing the 2013 review study¹⁸ were also selected.
4. Among the 78 selected articles, all the mentioned kinase-AE pairs were manually extracted.

We then examined the list of the updated kinase-associated toxicities following this literature survey effort against the model-identified KI-AE pairs shown in Figure 2a. We are happy to report that more predicated **KI-AE** pairs were found to be validated by the published experimental evidence and Figure 2a was updated accordingly. For instance, for diarrhea,

besides its well-established association with EGFR, the model has also identified its reported association with other kinases such as FLT3, VEGFR2 and AXL¹⁹.

Additionally, the results described above can also address what brought up by both reviewer #2 and #3: that is “no **new KI-AEs** were provided” and that “to demonstrate its utility, for example, the authors should move beyond validation of known associations and report (and discuss) one or two novel **kinase-AE** associations they have uncovered in the data.”

It must be noted that we first conducted predictions and obtained the results shown in original Figure 2a prior to the additional literature survey as described above following the review comments. Therefore, some KI-AE pairs “newly” identified via the ML model (i.e., yellow-colored AEs in Figure 2a) did not represent known associations at the time (i.e., not confirmed by the 2013 paper). Following the survey, we found many of “newly” identified pairs could be validated by experimental evidence, thus further support the credibility of the ML predictions. While there might be other model identified KI-AE pairs, those will not be able to be confirmed until new clinical evidence is published.

We added the description of the literature survey in the **Supplementary Information** section Literature survey on associations between kinase targets and AEs, revised the relevant paragraph in the *Results* section (please see lines 100-110) and updated Figure 2a.

The utility of Time to AE onset data in Table 1 is unclear to me, given that the range of values is so large (typically 1 day to ~ 2 years!). A more granular presentation of the data (for example a histogram of values), perhaps in supplemental data, would be welcome to show the distribution of the data.

Response:

To present the detailed distribution of time-to-events for each AE, Kaplan–Meier survival curves for the AE data sets were generated (Figure L5). The embedded table at the bottom part of the figure shows the cumulative numbers of events and the numbers at risk along the timeline. We added the figure to the **Supplementary Information** as Supplemental Figure S5.

In Figure 3, while model prediction of TEAEs was generally very good, it would be worth a deeper analysis of specific TEAEs that were not well predicted by the model. In particular, can the authors speculate why certain TEAEs with high observed frequency were not well predicted by the model?

Response:

As shown in Figure 3 in main text, the model predicted AEs with high probabilities ($\geq 10\%$) covered all the frequently reported TEAEs except the four AEs in the imatinib study shown below.

AE	Predicted probability by ML model	Observed incidence from clinical study
dyspepsia	0.09	0.12
hypophosphataemia	0.09	0.10
dysgeusia	0.05	0.10

weight increased	0.04	0.12
------------------	------	------

For AEs “dyspepsia” and “hypophosphataemia”, the ML model predicted probabilities fall marginally below the threshold 10%.

For “dysgeusia” and “weight increased”, we noticed that event occurrence rates of these two AEs are 5% and 2%, respectively, in the training dataset used for building ML models (particularly using RSF method). This level of sparsity in data could compromise the ML model performance, due to the high data dependency of ML approaches. Additionally, the distribution of AE onset date is significant different for “dysgeusia” between the training dataset and the validation clinical study data ($p < 0.01$, median onset date is 28 and 57, respectively). While RSF’s data-driven nature is great for handling the nonlinearity or high-dimensionality of the data, it may not efficiently extrapolate values that fall outside the training dataset. Thus, we speculate that lower AE event information contained in the training datasets affected ML model’s performance. We edited the *Results* section (line 159-163) to include the relevant speculation on the four AEs with predicted probability below 10%.

The web application the authors report is simple and user friendly. I can imagine some additional ways of interrogating the data that are not permitted in the current version that could be considered. For example, currently the data can be searched for a specific AE (with the results returning a ranked list of associated kinases). It could valuable to allow the reverse search, in which an investigator searches for a kinase of interest and the software would return a ranked list of associated AEs.

Response:

We thank the reviewer for the suggestion. As a result, we have launched another research effort and added the “Search by Kinase Target” module to the web platform <https://gongji.shinyapps.io/ml4ki/>. Searching a kinase will return a list containing all the associated AEs for which the kinase is listed as one of the top 25 VIMPs for the corresponding AE prediction. Of note, the VIMP values of kinases is specific to a certain AE. Therefore, VIMP values of a kinase are not to be compared and ranked directly across different AEs.

Figure L1 Architecture of a DeepHit model with two events for prediction. Reprinted from “DeepHit: A Deep Learning Approach to Survival Analysis with Competing Risks”, by Lee et al. *Proceedings of the AAAI Conference on Artificial Intelligence*, 32(1), p. 2316. Copyright c 2018, Association for the Advancement of Artificial Intelligence³.

Figure L2 Characteristics of the reporting odds ratio (ROR) and proportional reporting ratio (PRR) profiles for combined top ten AEs of vandetanib, afatinib, erlotinib, and nintedanib. The values of ROR and PRR are listed in the tables below for reference.

	abdominal pain	acne	alopecia	bronchitis	cough	decreased appetite	diarrhoea	dry skin	dyspnoea	fatigue	headache	hypertension	mucosal inflammation	nasopharyngitis	nausea	pruritus	rash	stomatitis	upper respiratory tract infection	vomiting
ROR																				
Vandetanib	2.078	1.199	0.875	0.434	0.529	0.824	0.450	0.875	0.720	1.399	3.380	6.763	0.076	0.712	1.034	0.696	0.695	0.190	0.778	0.777
Nintedanib	2.399	0.000	0.161	16.951	1.888	0.705	1.546	0.039	1.574	0.654	0.712	0.732	0.000	3.339	2.077	0.282	0.070	0.018	2.794	1.443
Erlotinib	0.372	1.194	3.868	0.087	1.763	1.584	0.667	1.046	3.817	2.218	0.677	0.160	0.354	0.128	0.982	2.121	3.060	0.540	0.385	0.584
Afatinib	0.260	2.383	1.198	0.055	0.632	1.203	1.131	3.001	0.373	0.752	0.484	0.229	21.863	0.510	0.498	1.750	1.813	10.536	0.536	0.975
PRR																				
Vandetanib	2.095	1.203	0.874	0.432	0.525	0.821	0.415	0.873	0.718	1.407	3.470	6.914	0.075	0.709	1.035	0.693	0.680	0.187	0.777	0.774
Nintedanib	2.420	0.000	0.160	17.348	1.910	0.700	1.650	0.038	1.581	0.650	0.709	0.730	0.000	3.414	2.131	0.279	0.065	0.018	2.830	1.454
Erlotinib	0.370	1.198	3.928	0.087	1.783	1.604	0.638	1.047	3.900	2.257	0.674	0.158	0.351	0.127	0.982	2.148	3.466	0.535	0.382	0.579
Afatinib	0.257	2.416	1.200	0.054	0.628	1.208	1.150	3.042	0.370	0.749	0.479	0.227	22.412	0.505	0.488	1.762	1.886	10.880	0.533	0.974

Figure L3 External validation on independent SMKI datasets: relationships between the model predicted AE probabilities and the observed AE incidences. Panels a, b and c are results based on two neratinib and one imatinib studies, respectively, corresponding to data used for generating Figure 3a, b and c in the manuscript. In each panel, the top row shows the scatterplots for the corresponding predicted probabilities vs all the frequently reported TEAEs (in $\geq 10\%$, $\geq 15\%$, $\geq 20\%$ of study subjects) and the bottom row figures for the corresponding observed TEAEs vs all AEs predicted to occur in $\geq 10\%$, $\geq 15\%$, $\geq 20\%$ of study subjects. Pearson's r is displayed on each plot. Yellow points represent frequently reported TEAEs with predicted probabilities below the selected threshold values (i.e., 10%, 15% or 20%); purple points AEs with both predicted probabilities and observed incidences equal or above the threshold values (i.e., 10%, 15%, or 20%).

Figure L4 Relationship between the model predicted AE probability and FAERS reported incidences. Sub-figures (a), (b) and (c) are results from two neratinib and one imatinib studies, respectively, corresponding to data used for generating Figure 3a, b and c in the manuscript. Spearman ρ is displayed on each plot.

(a) Neratinib Study I

(b) Neratinib Study II

(c) Imatinib study

Figure 2a Potential association between kinase and AE as identified by ML modeling. Variable importance (VIMP) assessment of predictive variables for five representative AEs. Bar length indicates the VIMP value of the variable, which represents the difference in the out-of-bag model prediction errors with and without this predictive variable being permuted²⁰. The identified top 25 variables are listed for each AE. The blue-highlighted kinases are the representative experimentally well-established KI-AE pairs¹⁸. The yellow-highlighted pairs are validated by KI-AE pairs found in literature survey results for KI-AE pairs post 2013 publication¹⁸ (Supplemental Table S3).

Figure L5 Kaplan–Meier survival curves for the patient population data for five representative AEs in Table 1.

Table L1 Predictive performances of the ML models on patient population data via C-index. The results are shown in mean of C-index (90% confidence interval). For each AE, the ML method with the best performance in terms of C-index is marked in bold.

AE (patients affected%)	DeepHit	ANN	RSF
Diarrhea (41.6)	0.651 (0.577, 0.724)	0.626 (0.524, 0.688)	0.712 (0.689, 0.735)
Acne (2.4)	0.790 (0.743, 0.836)	0.855 (0.781, 0.927)	0.852 (0.779, 0.912)
Hypertension (9.3)	0.832 (0.822, 0.841)	0.792 (0.642, 0.853)	0.782 (0.751, 0.809)
Conjunctivitis (2.2)	0.708 (0.607, 0.809)	0.685 (0.513, 0.815)	0.584 (0.510, 0.638)
Proteinuria (1.5)	0.815 (0.787, 0.843)	0.816 (0.609, 0.912)	0.784 (0.696, 0.868)
Pneumonia (2.5)	0.568 (0.509, 0.626)	0.594 (0.430, 0.741)	0.552 (0.445, 0.643)
Qt prolongation (2.5)	0.809 (0.770, 0.848)	0.793 (0.596, 0.891)	0.836 (0.776, 0.887)
Headache (14.2)	0.576 (0.558, 0.595)	0.554 (0.443, 0.619)	0.596 (0.554, 0.640)
Vomiting (17.0)	0.535 (0.420, 0.650)	0.694 (0.527, 0.748)	0.739 (0.699, 0.776)
Fatigue (19.3)	0.752 (0.740, 0.764)	0.697 (0.480, 0.756)	0.742 (0.711, 0.773)

Table L2 Search parameters used in U.S. FAERS queries for the safety profile regarding the two SMKIs neratinib and imatinib.

Drug product	neratinib	imatinib (RLD: Gleevec)
ANDA	None approved	Multiple (attached below)*
NDA Number	N208051	N021588
Product Number	001	002
Approval Date	Jul 17, 2017	Apr 18, 2003
Applicant Holder Full Name	Puma Biotechnology INC	Novartis Pharmaceuticals Corp
Initial FDA Received Date	07/17/2017 - 12/08/2021	04/18/2003 - 12/08/2021
MedDRA Version	24.1	24.1

Orange Book

* Approved Drug Produ

Table L3 Relationships between the model predicted AEs with high probability ($\geq 10\%$) and the top reported AEs in U.S. FAERS. Sub-tables (a) and (b) are results based on two neratinib studies and one imatinib study, respectively.

(a) Neratinib studies

Appear in model predicted AE with high probability ($\geq 10\%$)		Preferred Terms	Total Cases reported in U.S. FAERS	% of Cases reported in U.S. FAERS
Study I	Study II			
Yes		Diarrhea	824	61.80%
		Nausea	396	29.70%
		Fatigue	336	25.20%
		Constipation	206	15.50%
		Vomiting	184	13.80%
		Decreased appetite	158	11.90%
Not reported in the clinical trial*		Off label use	155	11.60%
		Death	128	9.60%
		Product dose omission issue	104	7.80%
5%	4%	Dehydration	88	6.60%
Yes		Abdominal pain	82	6.20%
		Abdominal pain upper	80	6.00%
		Headache	72	5.40%
Not reported in the clinical trial*		Hospitalisation	71	5.30%
Yes		Rash	66	5.00%
		Weight decreased	65	4.90%
7%	9%	Abdominal discomfort	64	4.80%
Yes		Dizziness	61	4.60%
		Dyspepsia	58	4.40%
6%	6%	Muscle spasms	56	4.20%

(b) Imatinib study

Appear in model predicted AE with high probability ($\geq 10\%$)	Preferred Terms	Total Cases reported in U.S. FAERS	% of Cases reported in U.S. FAERS
Not reported in the clinical trial*	Death	2,069	9.7%
Yes	Nausea	1,875	8.8%
Yes	Diarrhea	1,546	7.3%
Yes	Fatigue	1,235	5.8%
Not reported in the clinical trial*	Drug ineffective	1,093	5.1%
Yes	Vomiting	1,043	4.9%
Yes	Rash	934	4.4%
Yes	Muscle spasms	809	3.8%
Yes	Dyspnoea	674	3.2%
Not reported in the clinical trial*	Neoplasm malignant	621	2.9%
Yes	Asthenia	530	2.5%
Yes	Pain	488	2.3%
Not reported in the clinical trial*	Illness	485	2.3%
5%	Pneumonia	485	2.3%
Yes	Oedema peripheral	483	2.3%
4%	Malaise	470	2.2%
Yes	Anaemia	467	2.2%
Yes	Arthralgia	466	2.2%
Yes	Headache	433	2.0%
1%	Fluid retention	427	2.0%

* The preferred terms were not reported in the clinical trials of the 16 SMKIs used for training the predictive ML model, therefore no predictive ML model were built, and no prediction can be made for these AE preferred terms.

* The grey cells show top reported AEs in FAERS with model predicted probabilities of < 0.1

References

1. Gong, X., Hu, M. & Zhao, L. Big Data Toolsets to Pharmacometrics: Application of Machine Learning for Time-to-Event Analysis. *Clin Transl Sci* **11**, 305-311 (2018).
2. Lee, C., Yoon, J. & Schaar, M.V. Dynamic-Deephit: A Deep Learning Approach for Dynamic Survival Analysis with Competing Risks Based on Longitudinal Data. *IEEE Trans Biomed Eng* **67**, 122-133 (2020).
3. Lee, C., Zame, W., Yoon, J. & van der Schaar, M. Deephit: A Deep Learning Approach to Survival Analysis with Competing Risks *AAAI Conference on Artificial Intelligence Proceedings of the AAAI Conference on Artificial Intelligence*, 32(1). Retrieved from <https://ojs.aaai.org/index.php/AAAI/article/view/11842>
4. Biganzoli, E., Boracchi, P., Mariani, L. & Marubini, E. Feed Forward Neural Networks for the Analysis of Censored Survival Data: A Partial Logistic Regression Approach. *Statistics in Medicine* **17**, 1169-1186 (1998).
5. Faraggi, D. & Simon, R. A Neural Network Model for Survival Data. *Stat Med* **14**, 73-82 (1995).
6. Davis, M.I., Hunt, J.P., Herrgard, S. et al. Comprehensive Analysis of Kinase Inhibitor Selectivity. *Nat Biotechnol* **29**, 1046-1051 (2011).
7. Robichaux, J.P., Le, X., Vijayan, R.S.K. et al. Structure-Based Classification Predicts Drug Response in Egfr-Mutant Nscl. *Nature* **597**, 732-737 (2021).
8. Wood, E.R., Truesdale, A.T., McDonald, O.B. et al. A Unique Structure for Epidermal Growth Factor Receptor Bound to Gw572016 (Lapatinib): Relationships among Protein Conformation, Inhibitor Off-Rate, and Receptor Activity in Tumor Cells. *Cancer Res* **64**, 6652-6659 (2004).
9. Kumar, A., Petri, E.T., Halmos, B. & Boggon, T.J. Structure and Clinical Relevance of the Epidermal Growth Factor Receptor in Human Cancer. *J Clin Oncol* **26**, 1742-1751 (2008).
10. Maemondo, M., Inoue, A., Kobayashi, K. et al. Gefitinib or Chemotherapy for Non-Small-Cell Lung Cancer with Mutated Egfr. *N Engl J Med* **362**, 2380-2388 (2010).
11. Pao, W. & Chmielecki, J. Rational, Biologically Based Treatment of Egfr-Mutant Non-Small-Cell Lung Cancer. *Nat Rev Cancer* **10**, 760-774 (2010).
12. Pao, W., Miller, V.A., Politi, K.A. et al. Acquired Resistance of Lung Adenocarcinomas to Gefitinib or Erlotinib Is Associated with a Second Mutation in the Egfr Kinase Domain. *PLoS Med* **2**, e73 (2005).
13. Yang, X., Huang, Y., Crowson, M. et al. Kinase Inhibition-Related Adverse Events Predicted from in Vitro Kinome and Clinical Trial Data. *J Biomed Inform* **43**, 376-384 (2010).
14. Liu, J., Kim, G., Xu, J. et al. Combined Population Pk Modeling and Disproportionality Analyses to Assess the Association between Kinase Inhibition and Adverse Events. *117th ASCPT Annual Meeting San Diego, CA March 8, 2016*
15. Liu, J.Z., Kim, G., Xu, J. et al. Kinase Inhibitory Network Associated Side Effects (Kinase) Application. (2016) <https://jzliu.shinyapps.io/KINASE/>
16. Karaman, M.W., Herrgard, S., Treiber, D.K. et al. A Quantitative Analysis of Kinase Inhibitor Selectivity. *Nat Biotechnol* **26**, 127-132 (2008).
17. Federer, C., Yoo, M. & Tan, A.C. Big Data Mining and Adverse Event Pattern Analysis in Clinical Drug Trials. *Assay Drug Dev Technol* **14**, 557-566 (2016).
18. Dy, G.K. & Adjei, A.A. Understanding, Recognizing, and Managing Toxicities of Targeted Anticancer Therapies. *CA Cancer J Clin* **63**, 249-279 (2013).
19. Weitzman, S.P. & Cabanillas, M.E. The Treatment Landscape in Thyroid Cancer: A Focus on Cabozantinib. *Cancer Manag Res* **7**, 265-278 (2015).
20. Breiman, L. Random Forests. *Machine Learning* **45**, 5-32 (2001).

Reviewers' Comments:

Reviewer #1:

Remarks to the Author:

The authors have resolved my concerns.

Reviewer #2:

Remarks to the Author:

The authors have addressed my previous comments, and the revised manuscript demonstrated the utility of this research in linking kinase associated adverse event with kinase inhibitors via mining large-scale database.

Reviewer #3:

Remarks to the Author:

The authors have substantively addressed my concerns and I believe that the revised manuscript has been strengthened and will be of significant interest. As I suggested, the revised manuscript expands the authors' analysis to include additional KI-TEAE data beyond those originally reported in the 2013 study. The model performed well on this data and, further, identified previously unknown KI-TEAE associations. The new data has been incorporated into new and expanded figure panels. In addition, they have added a new feature to their web tool which allows users to search the data by kinase target. This will broaden the utility of the data by permitting interrogation either by TEAE or by a specific kinase. I have no further reservations about publication.